# Probing nanoscale oxygen ion motion in memristive systems

Yuchao Yang[1,*], Xiaoxian Zhang[2,3,*], Liang Qin[2,3,4], Qibin Zeng[2,3], Xiaohui Qiu[2,3] & Ru Huang[1]

Ion transport is an essential process for various applications including energy storage, sensing, display, memory and so on, however direct visualization of oxygen ion motion has been a challenging task, which lies in the fact that the normally used electron microscopy imaging mainly focuses on the mass attribute of ions. The lack of appropriate understandings and analytic approaches on oxygen ion motion has caused significant difficulties in disclosing the mechanism of oxides-based memristors. Here we show evidence of oxygen ion migration and accumulation in $HfO_2$ by *in situ* measurements of electrostatic force gradient between the probe and the sample, as systematically verified by the charge duration, oxygen gas eruption and controlled studies utilizing different electrolytes, field directions and environments. At higher voltages, oxygen-deficient nano-filaments are formed, as directly identified employing a $C_S$-corrected transmission electron microscope. This study could provide a generalized approach for probing ion motions at the nanoscale.

[1] Key Laboratory of Microelectronic Devices and Circuits (MOE), Institute of Microelectronics, Peking University, Beijing 100871, China. [2] CAS Key Laboratory of Standardization and Measurement for Nanotechnology, National Center for Nanoscience and Technology, Beijing 100190, China. [3] CAS Center for Excellence in Nanoscience, National Center for Nanoscience and Technology, Beijing 100190, China. [4] Academy of Advanced Interdisciplinary Studies, Peking University, Beijing 100871, China. * These authors contributed equally to this work. Correspondence and requests for materials should be addressed to Y.Y. (email: yuchaoyang@pku.edu.cn) or to X.Q. (email: xhqiu@nanoctr.cn) or to R.H. (email: ruhuang@pku.edu.cn).

on transport in solid-state materials is a fundamentally important process that moves charges simultaneously with mass transfer, enabling numerous applications from lithium batteries[1], solid-oxide fuel cells[2] to sensors[3], electrochromic displays[4] and lately, memristive devices[5–10]. Ion transport process is usually accompanied by electrochemical reactions, defect generations and phase transitions in the electrolytes or at the electrode/electrolyte interfaces, making it a highly complicated phenomenon that is interesting from both scientific and application perspectives. Memristive systems are an important class of nanoionic devices whose intriguing properties stem from fundamental cation (such as Ag and Cu ions)[11–13] or anion (for example, oxygen ions)[14–16] transport processes, leading to formation/dissolution of localized conduction channels and consequent evolution of resistance states. Since its theoretical inception in 1970s[17,18], followed by the establishment of connection with resistance switching behaviour in 2008 (ref. 19), memristive systems have attracted extensive research interests as a disruptive technology for nonvolatile memory[9,20,21], in-memory logic[22] and brain-inspired computing[23–25]. Depending on the moving ion species, memristive systems can

be further classified into electrochemical metallization memory (ECM) driven by the transport of metal cations[13,26] and valence change memory (VCM) where the resistance switching is presumably caused by the migration of oxygen ions/vacancies[9,27,28]. While the switching mechanism of ECM cells has been well understood, for example, through real time, *in situ* transmission electron microscopy (TEM) observations[11–13,29–31] as well as scanning tunnelling microscopy investigations[8], the mechanism of VCM devices and their internal ion transport dynamics still remain largely elusive, which may be ascribed to the low atomic number of oxygen ions and easy adsorption of oxygen contaminations from the ambient, imposing significant difficulties on acquiring unambiguous evidence regarding oxygen-deficient filaments and oxygen ion motion like that achieved in ECM cells. As a result, there is currently a pressing need for the characterization of anion transport and filament growth processes in VCM cells to guide continued device optimizations.

Fundamentally, the reason why resolving conducting filaments in VCM is challenging lies in the fact that TEM observations only focus on one of the two attributes related to ion transport—mass.

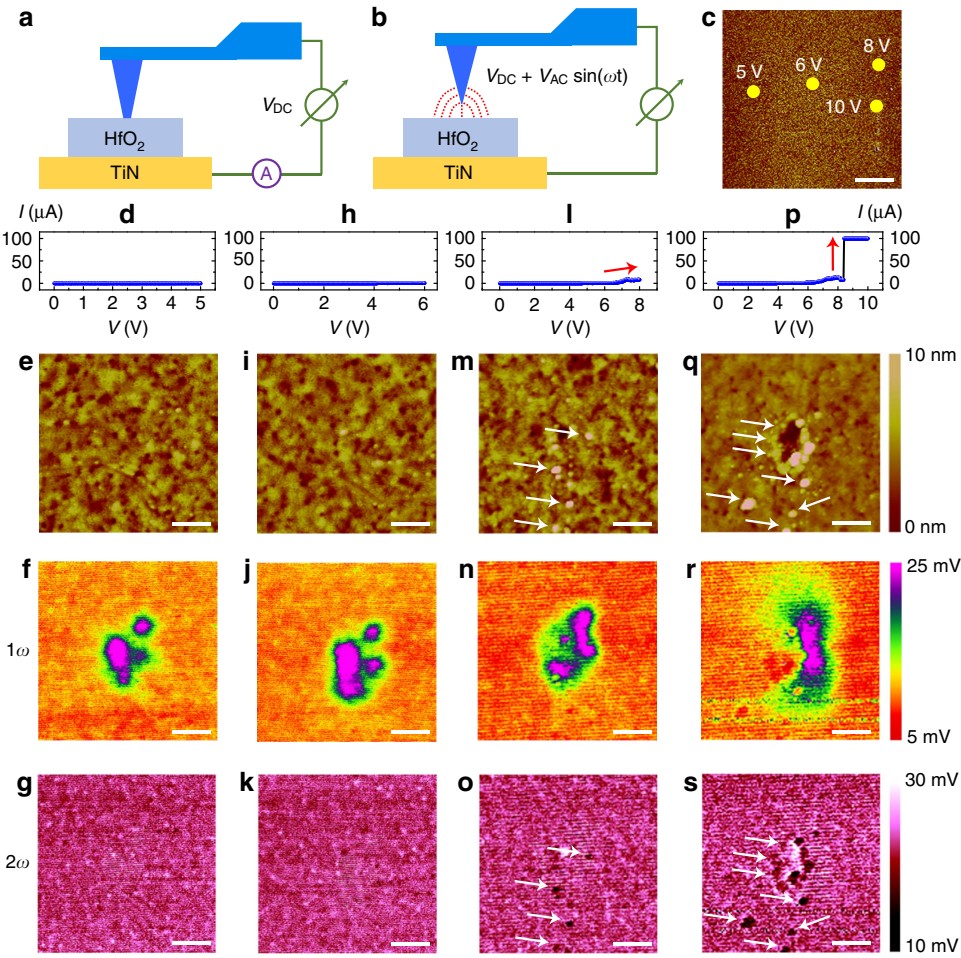

**Figure 1 | Imaging of dynamic ion accumulation during resistance switching process. (a)** Schematic of the conductive atomic force microscopy (C-AFM) measurements on HfO$_2$/TiN samples. **(b)** Schematic of the electrostatic force microscopy (EFM) measurements on HfO$_2$/TiN samples. **(c)** Topographic image showing the locations where voltage sweepings with different amplitudes were performed. Scale bar, 4 μm. **(d–g)** Electrical (**d**), topographic (**e**), 1$\omega$ (**f**) and 2$\omega$ (**g**) measurements on the region stimulated by voltage sweeping up to 5 V during preceding C-AFM measurements. Scale bar, 200 nm. **(h–k)** Corresponding electrical and EFM results on the region stimulated by voltage sweeping up to 6 V during preceding C-AFM measurements. Scale bar, 200 nm. **(l–o)** Corresponding electrical and EFM results on the region stimulated by voltage sweeping up to 8 V during preceding C-AFM measurements. Scale bar, 200 nm. **(p–s)** Corresponding electrical and EFM results on the region stimulated by voltage sweeping up to 10 V during preceding C-AFM measurements. Scale bar, 200 nm.

Instead, exploiting the charge attribute of the ions may lead to brand new opportunities in tackling this problem.

Here we show direct evidence of oxygen ion motion in HfO$_2$-based memristive devices by electrostatic force microscopy (EFM) that is sensitive to charge accumulations, possess sub-10 nm spatial resolution, and has no stringent requirements on the vacuum level or sample thickness[32,33]. Combined with systematic atomic force microscopy (AFM) and conductive atomic force microscopy (C-AFM) characterizations[34], this offers a reliable approach to probing ion transport processes. It was found that the migration and accumulation of oxygen ions can be detected by the electrostatic force between the probe and the sample, as has been systematically verified by the phase shift, charge duration as well as controlled studies utilizing different electrolyte materials, electric field directions and environments. When the oxygen ions move to the interfaces, subsequent redox reactions occur leading to oxygen gas eruption and oxygen-deficient filament formation. Such formation of conduction channels in HfO$_2$ is directly identified by high-resolution energy-dispersive X-ray spectroscopy (EDS) analysis utilizing a spherical aberration (C$_S$)-corrected TEM. The formation and dissolution of oxygen-deficient filaments, along with reversible ion motion experimentally visualized in the forming and reset processes, consistently explain the mechanism of bipolar resistance switching in HfO$_2$. These findings based on electrostatic and microstructural characterizations exploiting both the charge and mass attributes of ions could largely promote understandings on the switching mechanism of oxides-based memristors, and the present approach to probing ion transport dynamics in solid electrolytes may facilitate a wide variety of applications based on solid-state ionics, including energy storage, sensing, electrochromic display, computing and so on.

## Results

**Detection of ion motion in oxides.** To analyse oxygen ion migration in HfO$_2$, a HfO$_2$ (5 nm)/TiN sample was prepared on SiO$_2$/Si substrates, on which two successive steps of scanning probe microscopy (SPM) characterizations were performed, as shown in Fig. 1a,b. Firstly, the sample was probed by a Pt tip on the surface in contact mode to perform C-AFM measurements, and d.c. voltage sweeping was applied onto the tip with respect to the TiN electrode (see Methods for details), which reproduced typical device structures and operations of HfO$_2$-based memristive devices[35,36], as shown in Fig. 1a, where the Pt tip served as the top electrode. After electrically stimulating the HfO$_2$/TiN sample in different controlled conditions by varying the amplitude of the sweeping voltage (5, 6, 8 and 10 V, see Fig. 1c,d,h,l,p), dual-pass EFM measurements were performed to acquire topographic profiles of the stimulated areas in tapping mode in the first pass, followed by measurements of 1$\omega$ and 2$\omega$ components of the electrostatic force gradient between a Pt–Ir-coated silicon probe and the switching region by lifting the probe to a height of about 20 nm above the surface in the second pass (Fig. 1b), where $\omega$ represents the angular frequency of the applied a.c. voltage. One can see that in the HfO$_2$/TiN sample region following 5 V voltage sweep, pronounced changes in the 1$\omega$ signal can be clearly observed (Fig. 1f) even if the device still stays in high resistance state (Fig. 1d). In the meantime, no apparent features were seen in the topographic profile or the 2$\omega$ component, as shown in Fig. 1e,g, respectively.

In general, during the EFM scan (Fig. 1b) a small a.c. voltage ($V_{a.c.}$) at frequency $\omega$ is added to the d.c. bias voltage ($V_{d.c.}$) applied to the probe, and the total voltage difference between the probe and the sample would be

$$V_{tot} = \varphi + V_{d.c.} + V_{a.c.} \sin(\omega t) \qquad (1)$$

where $\varphi$ is the contact potential difference. The total electrostatic force can thus be formulated as follows[37]:

$$F_{EFM} = \frac{1}{2}\frac{dC}{dz}V_{tot}^2 + E_z Q_{tip} \qquad (2)$$

where $C$ is the capacitance between the probe and the sample, $z$ is the distance of the probe apex from the sample surface, $E_z$ is the $z$-component of the electric field at the tip position and $Q_{tip}$ is the charge on the tip. The first and second terms in equation (2) describe the contributions from capacitive forces due to surface potential and dielectric screening as well as Coulombic forces due to static charges and multipoles, respectively. Considering the fact that $E_z$ is contributed by electric field components due to static charges and multipoles $(E_z^S)$ as well as oscillating polarizations $(f(\varepsilon, \{g\})V_{a.c.}$, where $\varepsilon$ is the dielectric constant and $\{g\}$ denotes geometric parameters of the capacitor), while $Q_{tip}$ originates from both the capacitive charge ($CV_{tot}$) and the image charge ($Q_{im}$) caused by the static charge distribution on the surface, equation (2) can be expanded to the following form[37]:

$$F_{EFM} = \frac{1}{2}\frac{dC}{dz}\left((V_{d.c.}+\varphi)^2 + \frac{1}{2}V_{a.c.}^2\right) + E_z^S(Q_{im}+C(V_{d.c.}+\varphi))$$
$$+ \frac{1}{2}|f(\varepsilon,\{g\})V_{a.c.}|CV_{a.c.} + F(\omega)\sin(\omega t) - F(2\omega)\cos(2\omega t) \qquad (3)$$

with

$$F(\omega) = \left((V_{d.c.}+\varphi)\frac{dC}{dz} + E_z^S C + f(\varepsilon,\{g\})(Q_{im}+C(V_{d.c.}+\varphi))\right)V_{a.c.} \qquad (4)$$

and

$$F(2\omega) = \left(\frac{1}{2}f(\varepsilon,\{g\})C + \frac{1}{4}\frac{dC}{dz}\right)V_{a.c.}^2 \qquad (5)$$

One can see that the first three terms in equation (3) have no frequency dependence and are constants in fixed conditions, while the 1$\omega$ and 2$\omega$ components of the electrostatic force between the probe and the sample are described by the fourth and the fifth terms, respectively, whose amplitudes are formulated in equations (4) and (5) in detail. It should be noted that in our experiments we used phase detection that is sensitive to the gradient of the electrostatic force rather than the force itself to produce higher spatial resolution and more quantitative measurements[38], but the surface information revealed are nearly identical, depends on the same variables and can therefore be understood from the same set of equations. As shown in equation (5), the 2$\omega$ component is mainly influenced by the capacitance between the probe and the sample along with related capacitive parameters such as sample polarizability, while $z$ and $V_{a.c.}$ are decided by the experimental setup and conditions. As a result, features will appear in the 2$\omega$ scans when the dielectric constant $\varepsilon$ or the geometric parameters $\{g\}$ of the capacitor are altered (which in turn change $C$). The dielectric constant is expected to stay unchanged unless the phase or stoichiometry of HfO$_2$ varies, while the geometric parameters can also change, for example, upon structural deformations occurring to the stimulated region due to strong Joule heating effects or oxygen gas eruptions[14,39,40]. Since the topographic scan in Fig. 1e shows uniform surface profile of HfO$_2$ after C-AFM measurements, it is obvious that no structural deformations took place in the sample. Besides, the absence of pronounced features in the 2$\omega$ scan (Fig. 1g) suggests that the dielectric constant and thus the stoichiometry of the HfO$_2$ film also remains intact under such level of electrical stimulus.

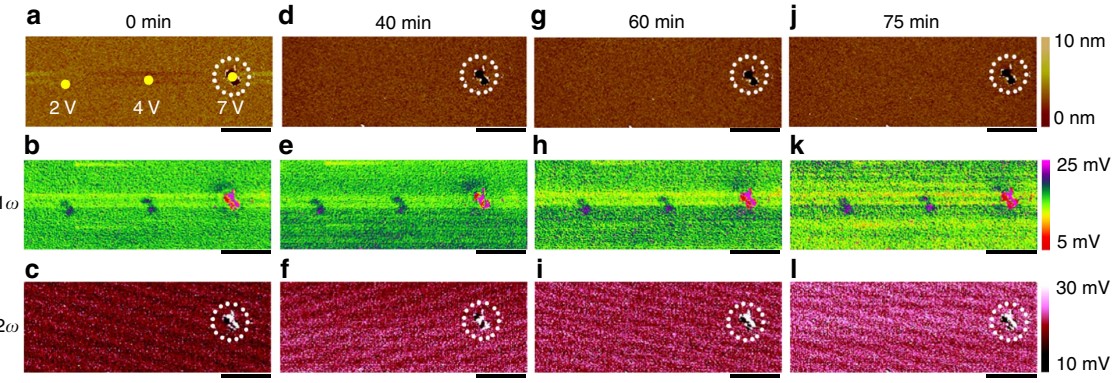

**Figure 2 | Retention test of the accumulated charges implying ionic nature.** Topographic and EFM results 0 min (**a–c**), 40 min (**d–f**), 60 min (**g–i**) and 75 min (**j–l**) after removing the electrical stimuli, showing long retention of the charges. Scale bar, 3 µm. This suggests that the charges observed in the 1$\omega$ component of the electrostatic force gradient should correspond to oxygen ions, instead of electrons.

Meanwhile, one can see from equation (4) that the 1$\omega$ component of the electrostatic force is related to testing parameters such as $V_{d.c.}$, $V_{a.c.}$ and material-related properties such as $\varphi$, which are unchanged under given conditions. Moreover, similar with $F(2\omega)$, the 1$\omega$ component of the electrostatic force is also affected by the capacitance and related capacitive parameters including $C$, $\varepsilon$ and $\{g\}$. Since we have experimentally confirmed that no features were observed in the 2$\omega$ scan in Fig. 1g, the contributions of $C$, $\varepsilon$ and $\{g\}$ to the 1$\omega$ feature as observed in Fig. 1f can be safely excluded in the present case. The mechanism of the 1$\omega$ pattern detected in Fig. 1f can therefore be attributed to the two remaining factors, that is, $Q_{im}$ and $E_z^S$, both of which point to charges that locally accumulate near the surface during the preceding C-AFM measurements. Since the probe was positively biased in the voltage sweeping, the accumulated charges should originate from either electrons trapped therein or negatively charged ions, that is, oxygen ions in the case of HfO$_2$. It should be noted that Valov and co-workers have shown that besides anions, cations can also move in HfO$_2$ (ref. 41), whereas in the present case the cation motion will be blocked as no reversible electrode (for example, Ta) exists in the analysed system. The negative sign of the charges was further verified by the attractive instead of repulsive force between a positively biased probe and the stimulated region, as demonstrated by the negative phase shift in EFM measurements (Supplementary Fig. 1; Supplementary Note 1)[38,42,43]. Similar observations were acquired when the sweeping voltage was increased to 6 V (Fig. 1h–k), except that the charged area expands (Fig. 1j), implying increased charge accumulation. When the applied voltage was further increased to 8 V, the conductance of the sample started to increase, as can be seen in Fig. 1l. Besides similar charge accumulations observed in the 1$\omega$ signal (Fig. 1n), a number of dark spots emerge in the 2$\omega$ component in the present case, as marked by the arrows in Fig. 1o. Based on previous analysis, these dark sports are indicative of variations in $\varepsilon$ or structural deformations to the sample that could alter the tip–sample capacitance. Indeed, such structural deformations are clearly visible in exactly the same locations in the topographic profile (Fig. 1m), as marked by the arrows. Such structural deformations have been reported to be a result of oxygen ion oxidation and subsequent oxygen gas formation at the top interface[14,39], following the process of

$$2O_O^{\times} \rightarrow O_2 + 2V_O^{\bullet\bullet} + 4e^- \qquad (6)$$

It implies that the accumulated charges shown in 1$\omega$ signal at lower voltages (Fig. 1f,j) should indeed correspond to oxygen anions, and in the present case of Fig. 1m–o with increased

voltages sufficient energy is gained to overcome the barrier of the redox reaction shown in equation (6), therefore leading to oxidation of the accumulated oxygen ions. Note, that the process in equation (6) will inevitably lead to local non-stoichiometry in the HfO$_2$ film, therefore the dielectric constant will also be affected ($\varepsilon \rightarrow \varepsilon'$), further accounting for the features observed in the 2$\omega$ signal (Fig. 1o). Such introduction of structural deformation and non-stoichiometry by enhanced electrical stimulus has been verified by further increasing the sweeping voltage to 10 V, where resistance switching from off to on state was clearly observed (Fig. 1p), implying non-stoichiometric filament formation typically with lower oxygen concentration[14–16]. Besides, growing structural damages to the HfO$_2$ film could also be visualized (Fig. 1q). Pronounced patterns were therefore observed in the 2$\omega$ signal (Fig. 1s) due to the non-stoichiometry and structural deformation, in light of the previous explanations.

Based on the above analysis, the 1$\omega$ and 2$\omega$ components of the electrostatic force gradient in the present setup consisting of memristive systems exhibit sensitivities to the charges that locally accumulate near the surface of the stimulated region and non-stoichiometry as well as structural distortions to the analysed sample, respectively. Dynamic processes that are involved during the resistance switching in terms of electrical, topographic and electrostatic characteristics have been successfully observed at the microscopic level (Fig. 1). Note that a positive voltage applied on the tip during C-AFM could induce the accumulation of either electrons or oxygen ions beneath the probe, as verified by the attractive electrostatic force (Supplementary Fig. 1; Supplementary Note 1). To further discriminate the nature of the accumulated charges, we have conducted retention tests, as shown in Fig. 2. It is expected that the accumulated charges should escape quickly if they correspond to trapped electrons, and typical emission time for electrons trapped in HfO$_2$ was widely reported to be around millisecond level[44–46]. On the contrary, in case of ions much longer retention should be observed before homogeneous ion distribution can be restored via backward diffusions due to their higher mass. Following this consideration, C-AFM measurements were performed on the HfO$_2$/TiN sample with voltage sweepings to 2, 4 and 7 V at different locations (Fig. 2a), and Fig. 2 shows the topographic and EFM measurements 0, 40, 60 and 75 min after removing the electrical stimuli. One can see that structural damages occurred to the sample when the sweeping voltage exceeded 7 V, while with lower voltages charge accumulations were once again observed in the electrostatic signal (Fig. 2b,e,h,k), in agreement with Fig. 1. Notably, the accumulated charges remain stable after over 75 min, as verified in Fig. 2k, which strongly suggests that the charges

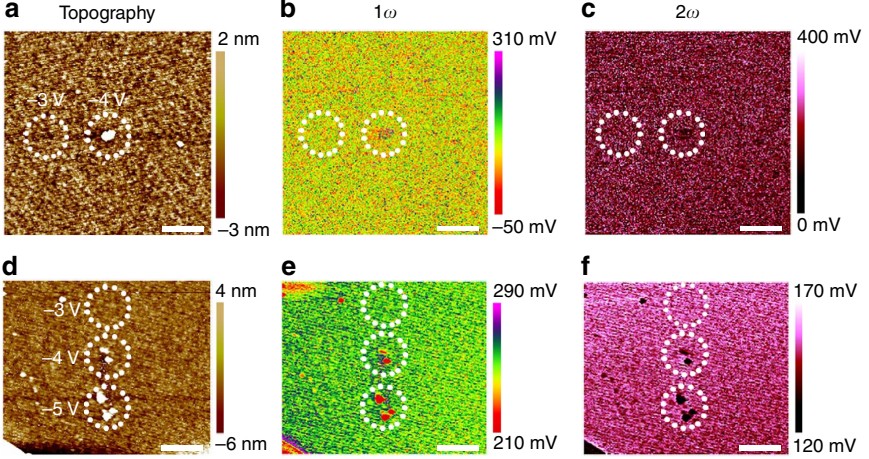

**Figure 3 | Electrostatic force microscopy characterization after negative voltage sweeps.** (**a–c**) Topographic (**a**), $1\omega$ (**b**) and $2\omega$ (**c**) measurements on the $HfO_2$/TiN sample undergoing negative voltage sweepings in preceding C-AFM measurements. The voltage amplitudes were –3 and –4 V at different locations. Scale bar, $1\,\mu m$. (**d–f**) Topographic (**d**), $1\omega$ (**e**) and $2\omega$ (**f**) measurements on the $HfO_2$/TiN sample undergoing negative voltage sweepings. The voltage amplitudes were –3, –4 and –5 V at different locations. Scale bar, $1\,\mu m$. The regions of interest are marked by circles. These repeated studies showed highly consistent results, where no charge accumulations were observed in case of negative voltage sweeps until structural deformations took place.

should correspond to oxygen ions instead of electrons. This significant departure in charge duration clarifies the ionic rather than electronic nature of the accumulated charges, making the electrostatic force a reliable probe for analysing oxygen ion transport dynamics in oxide memristors.

To further demonstrate this, we have also performed comparative EFM studies where negative voltage sweeps were applied in C-AFM experiments instead. We have conducted two independent measurements in Fig. 3a–f (Supplementary Fig. 2; Supplementary Note 2), and highly consistent results were obtained. In stark contrast to Figs 1 and 2, the oxygen ions in the present case are expected to be driven to the bottom $HfO_2$/TiN interface due to the reversed electric field direction, and hence the previously observed charge accumulations using EFM (Figs 1f,j,n,r and 2b,e,h,k) should no longer be visible since the oxygen vacancy concentration in as-prepared $HfO_2$ should be very low. Indeed, Fig. 3 shows that no charge accumulations have been observed in the $1\omega$ signal until structural distortions take place beyond –4 V, which once again demonstrates that the charges observed in the $1\omega$ component should indeed correspond to oxygen ions.

The validity of this correlation was once again corroborated by control experiments performed on $Al_2O_3$/TiN samples with a similar dielectric thickness (about 5 nm), as shown in Fig. 4. It is well known that $Al_2O_3$ has extremely low diffusion coefficient of oxygen ions even close to its melting point ($\sim 2,000\,°C$) and also has a high activation energy of about 6.5 eV for oxygen diffusion[47]. This explains the fact that $Al_2O_3$ has rarely been the material of choice for VCM devices. As a result, the oxygen ion motion in $Al_2O_3$ should be significantly retarded, thus providing a model system for investigating oxygen ion motion using EFM. We have electrically stimulated the $Al_2O_3$/TiN sample in a series of controlled conditions (3, 5, 7, 9, 11 and 13 V) using C-AFM, as shown in Fig. 4a (corresponding $I$–$V$ curves during C-AFM measurements can be found in Supplementary Fig. 3, also see Supplementary Note 3), and obviously no sign of charge accumulation was observed until structural deformations took place (Fig. 4b). The emergence of structural deformations above 11 V in the sample (as marked by the arrows in Fig. 4a) could be ascribed to the fact that the applied electric field has exceeded the breakdown field of $Al_2O_3$ (ref. 48). The absence of charge accumulation under normal stimulating voltages is in

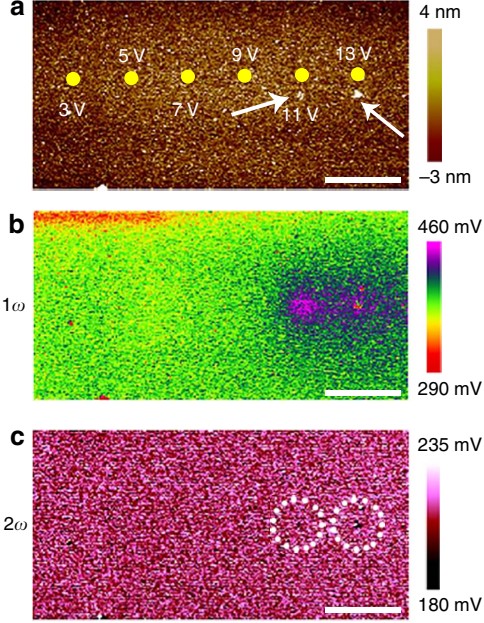

**Figure 4 | Electrostatic force microscopy characterization on $Al_2O_3$ with retardant oxygen ion migration.** (**a–c**) Topographic (**a**), $1\omega$ (**b**) and $2\omega$ (**c**) measurements on the $Al_2O_3$/TiN sample after voltage sweepings to 3, 5, 7, 9, 11 and 13 V at different locations. No charge accumulation was observed below 11 V. Scale bar, $4\,\mu m$. The arrows and circles indicate regions where structural distortions to the $Al_2O_3$ film were observed in the topographic profile and $2\omega$ signal, respectively.

stark contrast to the pronounced charge accumulation in $HfO_2$ at much lower voltage of 5 V, therefore consistent with the retardant oxygen ion diffusion in $Al_2O_3$. The above experiments have also ruled out the possibility of surface adsorbates, such as hydroxyl groups, serving as the origin of the accumulated charges, as otherwise it could not explain the absence of charge accumulation in $Al_2O_3$ given its similar or even higher hydroxyl site density compared with other oxides[49] under similar moisture levels during EFM testing. The influence of surface adsorbates on the EFM characterization was also evaluated by performing similar

C-AFM and EFM measurements in ultrahigh vacuum conditions ($\sim 8 \times 10^{-11}$ mbar). Highly consistent results with that in air were obtained, hence implying a subordinate role of random adsorbates from the ambient in EFM measurements (Supplementary Fig. 4; Supplementary Note 4). This once again demonstrates that the electrostatic force can indeed be reliably used as a probe for ion migrations in different electrolyte materials.

The probing depth of EFM, to first order, is roughly proportional to the product of the dielectric constant ($\varepsilon$) and the tip–sample distance ($z$), that is, $\varepsilon z$ (refs 50,51). Taking the dielectric constant of $HfO_2$ ($\sim 25$)[52] and the present experimental parameters ($z = 20$ nm) into consideration, the probing depth can theoretically reach about 500 nm. However, in practice, the thickness of the $HfO_2$ film is only 5 nm in the present study, and beneath $HfO_2$ the electric field will be fully screened by the metallic TiN electrode. As a result, in the present system the effective probing depth is limited by the dielectric thickness. It should be pointed out that although the reasonably large probing depth endows EFM with subsurface characterization capability[50], meaningful EFM signal will only be collected when electrostatic force arises, for example, upon local charge accumulations, as in

the case of Figs 1–2 and Supplementary Fig. 4. It is also worthwhile noting that the proposed approach is applicable to the characterization of ion motion in lateral devices as well in addition to the above devices in vertical configuration, if the capacitance effect can be safely decoupled (Supplementary Fig. 5; Supplementary Note 5).

**Resistance switching mechanism in VCM.** The above studies utilizing different electrolyte materials, field directions and time durations have unambiguously demonstrated the capability of EFM in facilely probing ion movements in a dynamic manner, which is desirable for understanding the switching mechanism of VCM devices. To do this, we have performed further C-AFM measurements on Ti/$HfO_2$/TiN samples. Compared with the sample structure in Figs 1–3, a Ti top electrode was inserted between the Pt tip and $HfO_2$/TiN. This structure has better resemblance with actual VCM cells (Supplementary Fig. 6a; Supplementary Note 6), hence allowing us to further study the impact of the abovementioned ion transport processes on practical memristive devices. The Ti/$HfO_2$/TiN memristive structure clearly display hysteretic current–voltage ($I$–$V$) characteristics in

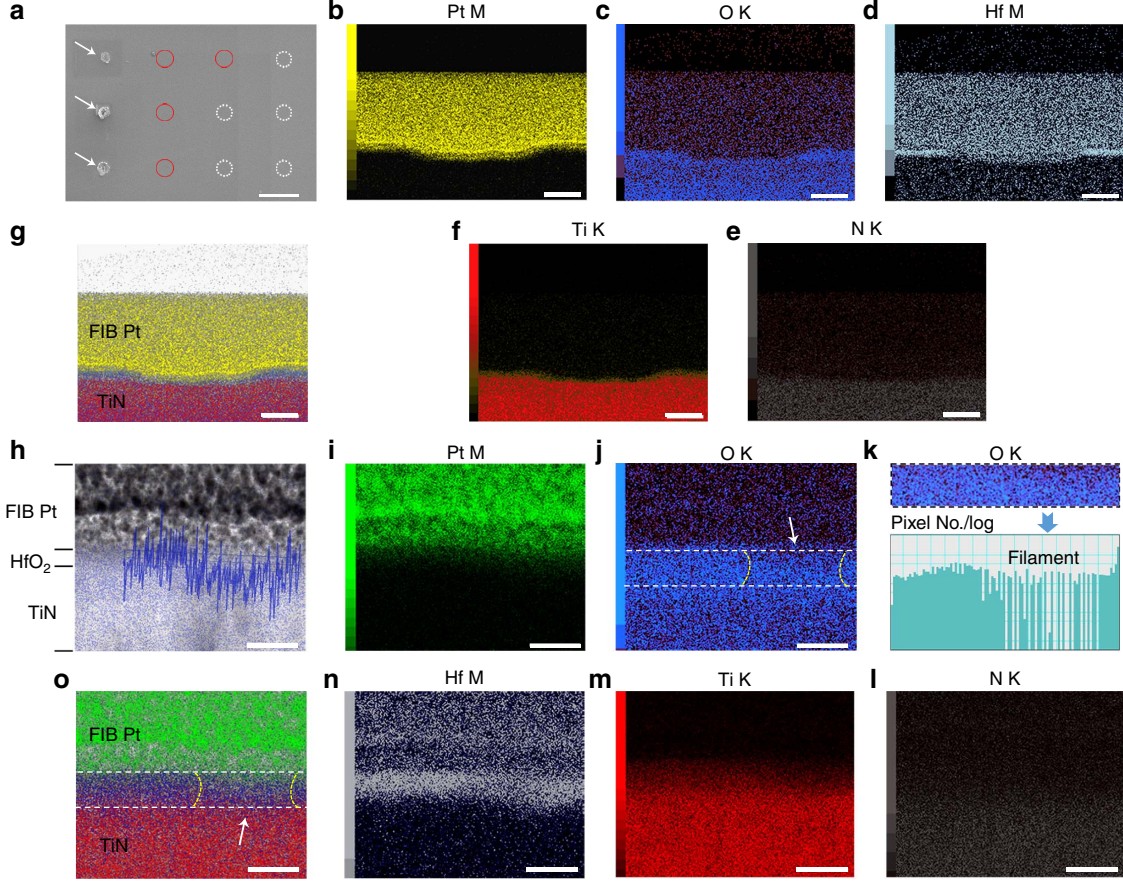

**Figure 5 | Direct identification of oxygen-deficient conducting filament in HfO₂.** (**a**) Scanning electron microscopy image showing a collection of electrically stimulated local regions using conductive atomic force microscopy. The regions were set into different controlled resistance states by applying varied programming conditions. The arrows indicate regions undergoing severe structural deformations by applying strong programming conditions, while the solid red and dotted white circles suggest regions existing at pristine and intermediate switching states by applying weak and moderate programming conditions, respectively. Scale bar, 2 μm. (**b**–**f**) Energy-dispersive X-ray spectroscopy (EDS) mapping of Pt M, O K, Hf M, Ti K and N K edges and (**g**) overlaid mapping results in a volcano region with structural deformations. Scale bar, 100 nm. (**h**) High-angle annular dark-field scanning transmission electron microscopy image overlaid with O K line profile and EDS mapping, showing formation of an oxygen-deficient filament in HfO₂. Scale bar, 10 nm. (**i**–**n**) EDS mapping of Pt M, O K, N K, Ti K, Hf M edges and (**o**) overlaid mapping results for the same region shown in **h**. The contour of the filament region was indicated by the dashed lines and the arrows. The histogram of O K edge signal in the HfO₂ film is also shown in **k**, highlighting the existence of an oxygen-deficient conduction channel on the right side of the HfO₂ film. Scale bar, 10 nm.

C-AFM measurements (Supplementary Fig. 6b), and a volcano-shaped bubble is formed on the Ti electrode after off→on switching (Supplementary Fig. 6c; Supplementary Note 6), which has been well understood to be a result of oxygen ions migration and their subsequent oxidations at the electrode/oxide interface via the process described by equation (6)[14,39]. This is once again consistent with the picture of oxygen ion migration shown by EFM, and it has been reported that persistent oxidations of oxygen ions at the interface will create oxygen vacancies in the oxide film, which is recognized as the driving mechanism for initial formation of oxygen-deficient filaments in oxides-based memrisitve devices[14,39,53]. This is exactly the case for Fig. 1p–s and Supplementary Fig. 6b,c, where the high conductance states observed from the I–V scans (Fig. 1p; Supplementary Fig. 6b) demonstrate that complete conducting filaments have already been formed in HfO$_2$, and the structural damages in the HfO$_2$/TiN sample (Fig. 1q) should be a result of gas eruption and enhanced Joule heating after the conduction channel is formed. The results in Fig. 1 and Supplementary Fig. 6a–c therefore point to a consistent picture during the resistance switching of HfO$_2$, where oxygen ions drift as driven by the electric field and the redox reactions at the interface lead to oxygen-deficient filament formation, structural deformation as well as oxygen gas eruption.

It is worthwhile pointing out that such oxygen ion motion and accompanying effects are universal phenomena for different oxides. C-AFM experiments on TaO$_x$/TiN samples have revealed very similar resistance switching and gas eruption characteristics (Supplementary Fig. 6d–f; Supplementary Note 6). It should also be noted that ambient conditions, in particular moisture levels, were found to play a significant role in filament formation (Supplementary Fig. 7; Supplementary Note 7), as verified by the easier programming of the same devices in air (with moisture) compared with that in high vacuum (with no or limited moisture). This is in agreement with previous studies on the effect of moisture in both VCM and ECM devices[54–56].

The abovementioned formation of oxygen-deficient filaments has been directly identified by detailed TEM and EDS characterizations using a C$_S$-corrected TEM instrument. To study resistance switching in HfO$_2$ at different stages, a number of varied switching conditions were applied on the HfO$_2$/TiN sample using C-AFM to create a collection of stimulated regions existing at different states, as shown by the scanning electron microscopy (SEM) image in Fig. 5a, where the arrows indicate volcano regions that have undergone strong programming conditions (programming current typically in the range of 4–100 μA, see Supplementary Fig. 8a,e,i, Supplementary Note 8) and hence showed obvious structural deformations to the sample, similar with the cases in Fig. 1q and Supplementary Fig. 6c,e,f. The solid red and dotted white circles suggest regions existing at pristine and intermediate switching states by applying relatively weak (typically 1–20 nA, as shown in Supplementary Fig. 8b,c,f,j) and moderate programming conditions (typically 50 nA–2 μA, as shown in Supplementary Fig. 8d,g,h,k,l), respectively. Following C-AFM measurements, these local regions were prepared into TEM specimens by focused ion beam (FIB) technique via a lift-out process (see Methods for details)[13]. The large number of switching regions (Fig. 5a) increases the probability for including the local conduction channels in the TEM specimen and thus offers a platform for understanding the physical mechanisms in different states. Once the TEM samples were ready, they were subjected to high-angle annular dark-field scanning transmission electron microscopy (HAADF STEM) imaging and detailed EDS analysis, and the time interval between the FIB preparation and STEM characterization were minimized to reduce oxygen contaminations from the ambient, to enhance the intrinsic signal. Figure 5b–f shows EDS mapping of Pt M, O K, Hf M, N K and Ti K edges from a volcano region that was successfully captured in the FIB specimen, and Fig. 5g further shows the overlaid mapping results. It is evident in Fig. 5b–g that the HfO$_2$ film in the volcano region was fully damaged under such strong programming conditions, corresponding to the cases in Fig. 1q as well as Supplementary Fig. 6c,e,f and implying strong thermal effects after filament formation. Such thermal effects lead to physical interactions between the probe and the sample as confirmed by comprehensive material characterizations, where it was found that the end of the Pt tip used in C-AFM can be melted during the volcano formation process (Supplementary Fig. 9; Supplementary Note 9). To get further insight into normal operation conditions of HfO$_2$-based VCM, detailed characterizations were also performed on regions existing at intermediate states by adopting moderate programming conditions, as shown in Fig. 5h–o, while results from the pristine state serve as a reference (Supplementary Fig. 10; Supplementary Note 10). Notably, there exist non-uniform distribution of oxygen on the right side of the HfO$_2$ film, as indicated by the dashed lines and the arrows in Fig. 5j,o, suggesting formation of a conducting filament therein. This is further confirmed by the histogram of O K edge signal in the HfO$_2$ film as shown in Fig. 5k, highlighting the existence of an oxygen-deficient conduction channel on the right side of the region. Such non-uniformity in composition is pronounced if compared with the pristine state of the sample, where overall uniform oxygen distribution can be observed (Supplementary Fig. 10), therefore excluding the possibility of experimental artifacts. We have further performed quantitative O K edge linescan to verify this (Fig. 5h, overlaid with the HAADF STEM image), and a local dip in oxygen concentration was once again experimentally observed in the suspected region, once again confirming the formation of an oxygen-deficient conduction channel in HfO$_2$. These results are fully consistent with the implications from the EFM, C-AFM and electrical results (Fig. 1; Supplementary Fig. 6), therefore underpinning the physical picture comprised of oxygen ion migration, redox reaction and subsequent filament formation in HfO$_2$-based memristors.

The above combination of EFM and TEM characterizations, exploiting the charge and mass attributes of ions respectively, have collectively illustrated the ion dynamics and resultant conducting filament formation in oxide based memristors. To further shed light on the reset mechanism, we conducted measurements using reversed electrical stimuli, as shown in Fig. 6a–c. We have firstly performed electrical stimulations on the HfO$_2$/TiN sample using a series of positive voltage sweeps (3, 5, 7 and 9 V, upper row of Fig. 6a) to retrieve the forming threshold, and one can see that structural distortions took place when the sweeping voltage exceeded 7 V, hence suggesting 5 V as an appropriate value that is approximate to the forming voltage. Indeed, corresponding electrical data show that the device current has reached a high level of about 0.2 μA at 5 V (Supplementary Fig. 11; Supplementary Note 11), therefore confirming the onset of filament formation. As a result, in the second run of experiments (bottom part of Fig. 6a–c) the HfO$_2$/TiN sample was firstly stimulated by 5 V voltage sweeps, which attract oxygen ions to the top interface. Afterwards, gradually increased negative stimulations were applied on the same locations (0, −1, −3 and −5 V) as depicted in the lower row in Fig. 6a. One can see that the oxygen ions originally attracted to the top interface gradually disappeared as the amplitude of the negative voltage increased (Fig. 6b). These results clearly demonstrate the dynamic ion migration processes involved during the bipolar switching, as schematically depicted in Fig. 6d–i. When a positive voltage sweep is applied, oxygen ions will be attracted to the top interface,

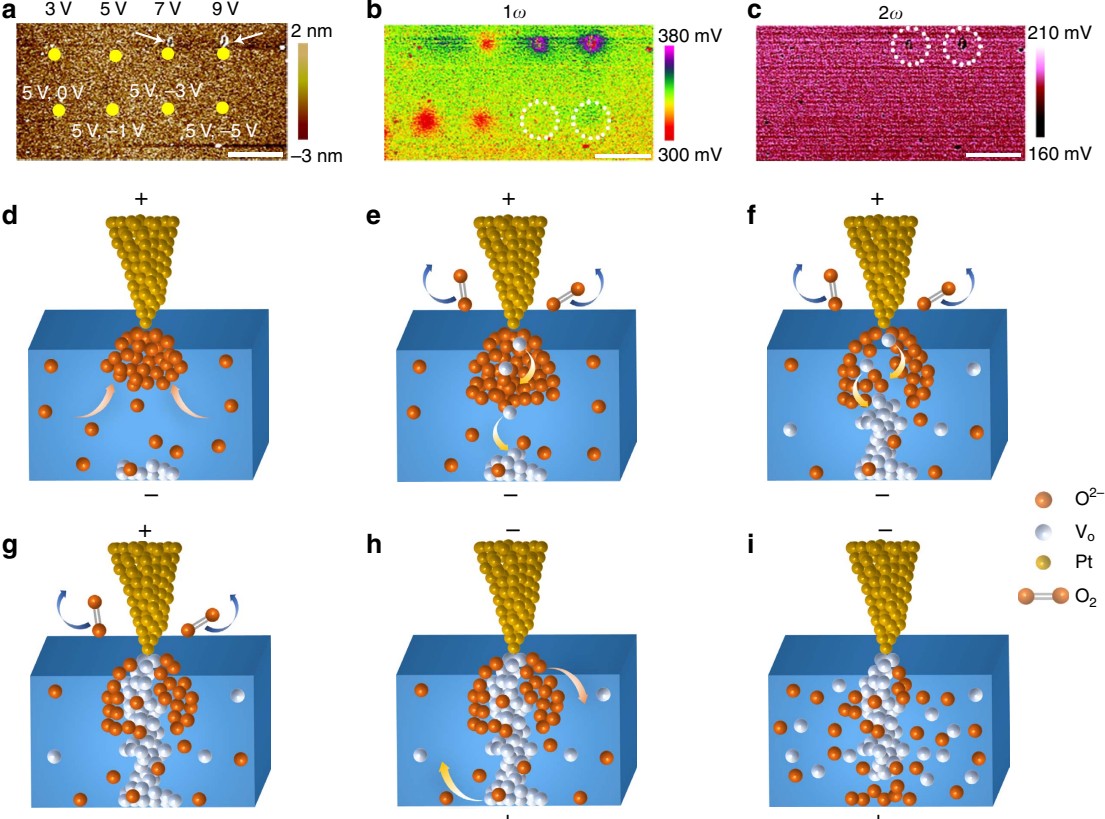

**Figure 6 | Reversible oxygen ion dynamics in forming and reset processes.** (**a–c**) Topographic (**a**), $1\omega$ (**b**) and $2\omega$ (**c**) measurements on the $HfO_2$/TiN sample after positive voltage sweepings to 3, 5, 7 and 9 V at different locations in the upper row, along with bidirectional voltage sweepings of (5 V, 0 V), (5 V, −1 V), (5 V, −3 V) and (5 V, −5 V) in the lower row. Scale bar, 3 μm. (**d–i**) Schematic illustration of dynamic ion motion and filament formation/ dissolution processes during the forming and reset processes of $HfO_2$ memristors. (**d**) A positive voltage on the Pt tip attracts oxygen ions to the top interface and repels oxygen vacancies to the bottom interface. (**e**) Oxidation of the oxygen ions at the top interface leads to oxygen gas eruption and oxygen vacancies formation. (**f**) The migration of oxygen vacancies to the bottom interface and continued redox reactions lead to gradual growth of the oxygen vacancy filament. (**g**) Formation of an oxygen-deficient filament through the $HfO_2$ film. The device is switched to on state in this case. (**h**) Reversal of applied voltage will drive oxygen ions away from the top interface and oxygen vacancies away from the bottom interface. (**i**) Continued application of negative voltage on the tip leads to dissolution of the oxygen-deficient filament. The device is switched to off state in this case.

while oxygen vacancies originally existing in the as-prepared $HfO_2$ film will get accumulated at the bottom interface (Fig. 6d). Subsequent oxidations of the oxygen ions at the top interface lead to eruption of oxygen gas to the ambient and formation of oxygen vacancies in the $HfO_2$ film (Fig. 6e), which has been demonstrated in Supplementary Fig. 6. Since the oxygen vacancies are positively charged, they will be driven toward the bottom interface, while additional oxygen ions will be attracted to the top interface and new oxygen vacancies will be persistently formed through the process described by equation (6), as illustrated in Fig. 6f, leading to gradual oxygen-deficient filament growth from the cathode. This eventually leads to formation of a complete conducting filament in $HfO_2$ film (Fig. 6g), as has been demonstrated by systematic STEM and EDS characterizations (Fig. 5). When the applied voltage is reversed, the oxygen ions originally attracted to the top interface will be driven downwards (as confirmed in Fig. 6b), while on the contrary the oxygen vacancies will be repelled from the bottom interface (Fig. 6h). This leads to dissolution of the conducting filament, and as a result the device is switched back to off state (Fig. 6i). Our comprehensive studies using a series of SPM and TEM characterizations with complementary capabilities have therefore elucidated the dynamic switching processes in $HfO_2$-based memristive devices, and the capability of resolving reversible and dynamic ion motion using *in situ* EFM provides a generalized approach for investigations of detailed ion transport in various memristive systems and electrolyte materials.

## Discussion

To summarize, we have shown direct evidence of oxygen ion motion in $HfO_2$- and $TaO_x$-based memristive devices by EFM measurements that focus on the charge attribute of ions, therefore providing a route to investigating ion transport dynamics in solid electrolytes. The migration of oxygen ions to the interface has induced redox reactions and oxygen gas eruptions, leading to oxygen-deficient filament formation and structural distortions in the electrolyte films. Such formation of conduction channels in $HfO_2$ was directly identified by high-resolution STEM and EDS characterizations utilizing a $C_S$-corrected TEM, and reversible ion motion has been found responsible for the forming and reset processes in $HfO_2$ memristors. These observations have been supported by comprehensive investigations using different electrolyte materials, field directions and time durations. The complementary capabilities of electrostatic (EFM) and micro-structural (TEM) characterizations, focusing on the charge and mass attributes of ions respectively, have collectively illustrated the ion motion and filament formation in oxide based memristors. We expect these findings will be of general significance not only for understanding the mechanism of

oxides-based memristive devices, but also for the detection and investigation of ion transport processes in solid electrolytes in general.

## Methods

**In situ EFM characterization.** The in situ EFM measurements were performed using a multimode AFM (Bruker Dimension Icon). The sample was firstly probed by a Pt tip (Bruker, RMN-12Pt400B) on the surface in contact mode to perform C-AFM measurements, where d.c. voltage sweeping was applied onto the tip with respect to the TiN electrode. The cantilever used in C-AFM has a resonant frequency of 4.5 kHz, with a spring constant of 0.3 N m$^{-1}$. After d.c. voltage sweeps, dual-pass EFM measurements were conducted in situ in tapping and lift mode using Pt–Ir-coated silicon tips (Bruker, SCM-PIT) with a resonant frequency and spring constant of 75 kHz and 2.8 N m$^{-1}$, respectively. The topographic profile was firstly obtained in the tapping mode without any voltage applied to the tip, followed by measurements of the $1\omega$ and $2\omega$ components of the electrostatic force gradient by lifting the probe to a height of about 20 nm above the sample, where 2 V d.c. offset with a modulating sinusoidal voltage was applied to the tip. The phase detection in d.c. mode EFM was conducted in lift mode with $+2$ V tip bias, and the bottom electrode of the sample was kept grounded in all of our EFM experiments. All the experiments in ambient were performed under similar temperature and moisture levels. The EFM and C-AFM studies were also performed in vacuum conditions using a Scienta Omicron variable temperature SPM system with an ultrahigh vacuum level of about $8 \times 10^{-11}$ mbar. A baking step at around 140 °C prior to the characterizations was performed to further remove the absorbed moisture from the sample.

**Microstructural and compositional characterization.** The TEM samples in this work were prepared from the locally stimulated regions of HfO$_2$/TiN sample by FIB technique using a dual-beam FIB system (FEI Helios Nanolab workstation). The HfO$_2$/TiN samples were stimulated in varied conditions to achieve different resistance states. During FIB patterning, the sample was firstly coated by a Pt layer deposited using the electron beam to avoid surface damages, followed by higher-rate Pt coating using normal ion beam process that serves as the majority of the protective layer during FIB cutting. In the end of the sample thinning, low-energy Ga$^+$ ion milling with extreme care was always used to obtain high-quality and thin TEM specimens. The specimens were then subjected to TEM, HAADF STEM and EDS characterizations using a C$_S$-corrected TEM instrument (JEM ARM200F) operated at 200 kV. The EDS measurements were performed in the STEM mode to obtain high spatial resolution, and a high-efficiency EDS detector was used during the analysis. The intensity in EDS mapping was analysed by the DigitalMicrograph software (Gatan Inc.) to get statistical histograms. The SEM characterization was conducted on the dual-beam FEI Helios Nanolab workstation and a field emission SEM (Hitachi S-4800). The Electron Probe Microanalyzer (EPMA) and wavelength dispersive X-ray spectroscopy (WDS) characterizations were performed on JEOL JXA-8230.

**Sample preparation.** Starting from Si substrates with 300 nm thermal oxides, global TiN electrode with a thickness of 200 nm was deposited by magnetron sputtering at room temperature. Subsequently, HfO$_2$ films with a thickness of 5 nm were deposited by atomic layer deposition (ALD) at 300 °C to form a HfO$_2$/TiN structure. In case of Ti/HfO$_2$/TiN samples with top electrodes, the Ti electrodes were patterned by electron beam lithography, magnetron sputtering and lift-off processes. The Ti electrodes have a thickness of 15 nm. The lateral devices with a gap of about 150 nm were prepared by electron beam lithography, electron beam evaporation and lift-off processes, where a 5 nm thick Ti layer was introduced beneath Pt electrodes (20 nm) as an adhesion layer. To form the TaO$_x$/TiN and Al$_2$O$_3$/TiN structures, TaO$_x$ film with a thickness of 30 nm was deposited by reactive sputtering with a Ar:O$_2$ ratio of 1:1 and RF power of 500 W at room temperature, while Al$_2$O$_3$ film with a thickness of 5 nm was deposited by ALD at 200 °C.

**Data availability.** All data supporting this study and its findings are available within the article, its Supplementary Information and associated files. Any source data deemed relevant is available from the corresponding author upon request.

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

## Acknowledgements

This work was supported by Beijing Municipal Science & Technology Commission Program (Z161100000216148) and National Natural Science Foundation of China (61674006, 61421005, 61376087, 61574007, 11604064, 21425310). Y.Y. acknowledges support from the '1000 Youth Talents Program' of China.

## Author contributions

Y.Y. conceived the study and prepared the manuscript. Y.Y., X.Z., X.Q. and R.H. directed all experimental research and constructed the research frame. X.Z., L.Q. and Q.Z. performed the C-AFM and EFM measurements. Y.Y. prepared all samples and performed the TEM, EDS, SEM, EPMA and WDS analyses. All authors analysed the results and implications and commented on the manuscript at all stages.

## Additional information

**Competing interests:** The authors declare no competing financial interests.



