## [Peer Review File · Nature Communications]

Reviewers' comments:

Reviewer #1 (Remarks to the Author):

The authors present direct visualization of ion motion and transport dynamics in metal-oxide based memristive material systems by utilizing an electrostatic force microscopy (EFM). Such ion motion was further supported by charge endurance test, TEM imaging of the channel cross section, and the observation of oxygen eruption feature (after the "hard" electroforming process). The authors finally proposed a model for forming and reset processes based on such EFM studies on "soft" electroformed and "off" state samples. The manuscript was clearly written.

In my view, the novelty of this work lies in providing a neat approach to visualize ion movement of "soft" electroformed solid electrolytes, which could work as an important complimentary technique to investigate microscopic pictures of ions by comparing to the popular but complicated in-situ TEM measurements. I think the manuscript is in a good shape in general, and after a few comments below are addressed, it could be suitable for publication in Nature Communications.

Major comments:

1. The first aspect is the environment chosen for the EFM measurements. In ambient air, random adsorption could bring significant effects on such surface-sensitive techniques. This is typically enhanced when electric field and electric-chemical reactions get involved. This requires careful investigation and detailed discussions.

2. Can the authors comment on the effective probing depth of this technique? I'll be curious to know if such technique still works on lateral thin film devices (other than vertical sandwich devices)? It will definitely bring higher impact to the field if it still works, and can be used to visualize the dynamics of the whole conduction channel for both set and reset processes.

Some minor questions:

1. For the O₂ bubble part (Fig. 5), since it has been previously reported and well understood by the community, I'll suggest to move this part to the supplementary information.
2. There are few typos in the manuscript. For example, on Page 8, "Fig. 11" should be "Fig. 1i".

Reviewer #2 (Remarks to the Author):

The manuscript by Yuchao Yang et al. reports on AFM-based studies combine with dielectric spectroscopy measurements and TEM/EDX analysis of VCM cells. HfO₂, Ta₂O₅ and Al₂O₃ oxides were used as variety of the solid electrolyte materials with different transport properties. The authors combined and amended several techniques to verify the oxygen ion dynamics within the studied oxides. I find particularly interesting the way of distinguishing oxygen ions from electrons by electrostatic measurements.

The manuscript is well structured and written. The authors suggest a nice idea and practically demonstrated its application. The topic is of principle interest and wroth of publishing in Nature Communications.

I have however, some comments needing to be addressed prior to a decision on the manuscript.

Comments:

1. The authors have confirmed that oxygen ions are attracted to the surface of the oxide at

positive bias. This implies that oxygen vacancies are attracted at the other (negative) electrode.

Reaching particular voltage, oxygen molecules will be formed at the positive electrode (as also discussed by the authors). The formed vacancies will diffuse to the cathode and new oxygen ions will come to be oxidized at the anode. If the vacancies cannot move away the oxygen ion transport (and therefore further reaction) will also be stopped.

The counter electrode reaction should be reduction of the oxide (oxygen vacancies + electrons) at the cathode. This is a classical situation known from the solid state ion conductors, where a virtual cathode (as defined for the VCM cells) is formed from a reduced oxide.

However, Fig 7 shows some inconsistency with this picture. Fig 7d shows (correctly) the accumulation of oxygen ions, but no accumulation of oxygen vacancies at the negative electrode is shown. Further Fig 7 (e-g) shows that the filament (reduced oxide) grows from the anode, whereas it should grow from the cathode (to form the filament as a virtual cathode). This point should be corrected in the figure.

2. The authors observed also morphological changes (craters) beneath the AFM-tip at higher voltages. Can the authors exclude some interaction (maybe alloying) between the tip and the oxide at these extreme conditions?

3. The electrical measurements have been performed at ambient conditions. Could the authors exclude effects from moisture? There are several reports on effects of moisture on the switching behavior of Ta₂O₅ and Al₂O₃.

4. Figure 6 should be improved. It is difficult to justify the results now. From the shown images it is almost impossible to distinguish any particular feature in the element distribution analysis. The size of the images should be enlarged.

We would like to greatly thank the reviewers for their very constructive suggestions, which we found valuable in improving our manuscript. We have carefully considered all the reviewers' questions, fabricated new lateral devices, performed additional measurements and made corresponding revisions. The point-to-point responses and the changes made are listed below.

Comments from Reviewer #1

Overall Remarks: The authors present direct visualization of ion motion and transport dynamics in metal-oxide based memristive material systems by utilizing an electrostatic force microscopy (EFM). Such ion motion was further supported by charge endurance test, TEM imaging of the channel cross section, and the observation of oxygen eruption feature (after the “hard” electroforming process). The authors finally proposed a model for forming and reset processes based on such EFM studies on “soft” electroformed and “off” state samples. The manuscript was clearly written.

In my view, the novelty of this work lies in providing a neat approach to visualize ion movement of “soft” electroformed solid electrolytes, which could work as an important complimentary technique to investigate microscopic pictures of ions by comparing to the popular but complicated in-situ TEM measurements. I think the manuscript is in a good shape in general, and after a few comments below are addressed, it could be suitable for publication in Nature Communications.

Our response: We would like to sincerely thank the reviewer for his/her positive comments which summarize the key findings of our work, and also for the reviewer's evaluation on the significance of the approach proposed here in serving as *“an important complimentary*

technique to investigate microscopic pictures of ions". We also deeply appreciate the valuable comments that this reviewer made and have carefully addressed all the points, as shown below.

1. The first aspect is the environment chosen for the EFM measurements. In ambient air, random adsorption could bring significant effects on such surface-sensitive techniques. This is typically enhanced when electric field and electric-chemical reactions get involved. This requires careful investigation and detailed discussions.

Our response: We would like to thank the reviewer for the valuable remark. The reviewer was asking about the influence of the environment, i.e. the ambient air, on the characterization of ion transport by EFM. *In order to assess the role of the environment especially surface adsorptions, we have adopted a new ultrahigh vacuum SPM setup (Scienta Omicron VT-SPM, vacuum level of $\sim 8 \times 10^{-11}$ mbar, shown in Supplementary Fig. 7a), and performed identical C-AFM and EFM measurements with that in air.* The electrical and EFM results are summarized in Supplementary Fig. 4, as also appended below for the reviewer's convenience.

One can see that the 1ω (Supplementary Figs. 4b,e,h,k) and 2ω (Supplementary Figs. 4c,f,i,l) components of the EFM signal between the probe and the switching region after applying voltage sweeps during the preceding C-AFM measurements (Supplementary Figs. 4a,d,g,j) showed highly consistent results with that in air (Fig. 1), therefore demonstrating the subordinate role of random adsorbates from the ambient in EFM characterization. It should be pointed out that in this new ultrahigh vacuum SPM setup, we used a metallic multi-walled carbon nanotube with much smaller diameter as the probe instead of the conventional Pt-Ir coated silicon tip, which accounts for the smaller area of the charge-accumulation regions (Supplementary Figs. 4b,e,h,k). The 1ω signal is otherwise very similar. In addition, it is well known that the resistance switching process itself (instead of the EFM measurement) will be retarded in vacuum condition, since the moisture in ambient air plays a role in facilitating the redox reactions and charge transfer processes that are indispensable for resistive switching (Refs. R1–R4). As a result, the switching voltage was increased to ~ 10 V in vacuum (compared with ~ 8.4 V in air), and the current level was also much lower due to the current compliance of the new equipment (~ 200 nA, compared with $100 \mu\text{A}$ in Fig. 1p), as shown in Supplementary Fig. 4j. The thermal effect during switching was thus significantly reduced, which in turn

avoids severe structural damages shown in Fig. 1q and hence prevents pronounced features from being observed in the 2ω signal. All the results are highly consistent with that in ambient conditions and precludes the influence of surface adsorptions.

It is also worthwhile noting that our EFM studies on the Al_2O_3 system with extremely low oxygen ion diffusion coefficient can also help rule out the influence of surface adsorbates on EFM measurements (Fig. 4). Since the measurements on HfO_2 and Al_2O_3 were performed under the same ambient conditions, similar charge accumulation behavior with that in HfO_2 should be observed in Al_2O_3 if there exists significant impact from the environment. However, no sign of charge accumulation was observed in Al_2O_3 until structural deformations took place, which could only be ascribed to the intrinsic ion transport properties.

In order to clarify this point in the manuscript, we have added the following discussions in Page 12 of the revised manuscript: *“The influence of surface adsorbates on the EFM characterization was also evaluated by performing similar C-AFM and EFM measurements in ultrahigh vacuum conditions ($\sim 8 \times 10^{-11}$ mbar). Highly consistent results with that in air were obtained, hence implying a subordinate role of random adsorbates from the ambient in EFM measurements (see Supplementary Fig. 4, Supplementary Note 4).”*

Supplementary Figure 4 | C-AFM and EFM measurements in ultrahigh vacuum

condition. (a–c) Electrical (a), 1ω (b) and 2ω (c) measurements on the region stimulated by voltage sweeping up to 5 V during preceding C-AFM measurements. **(d–f)** Corresponding electrical and EFM results on the region stimulated by voltage sweeping up to 6 V during preceding C-AFM measurements. **(g–i)** Corresponding electrical and EFM results on the region stimulated by voltage sweeping up to 8 V during preceding C-AFM measurements. **(j–l)** Corresponding electrical and EFM results on the region stimulated by voltage sweeping up to 10 V during preceding C-AFM measurements.

2. Can the authors comment on the effective probing depth of this technique? I'll be curious to know if such technique still works on lateral thin film devices (other than vertical sandwich devices)? It will definitely bring higher impact to the field if it still works, and can be used to visualize the dynamics of the whole conduction channel for both set and reset processes.

Our response: We would like to greatly thank the reviewer for raising these interesting points.

The first part of the question is regarding the probing depth of EFM. To first order, this is proportional to the product of the dielectric constant (ϵ) and the tip-sample distance (z), i.e. $\sim\epsilon z$, based on the uniform-line-charge model (Refs. R5–R6). Taking the dielectric constant of HfO_2 (~ 25) (Ref. R7) and the present experimental conditions ($z = 20$ nm) into consideration, the probing depth can theoretically reach ~ 500 nm. However, in reality, the thickness of the HfO_2 film is only ~ 5 nm in the present study, and beneath that the electric field will be fully screened by the TiN electrode. As a result, herein the effective probing depth is limited by the thickness of the oxide instead of $\sim\epsilon z$. It is also worthwhile noting that meaningful EFM signal can only be detected when electrostatic force arises, e.g. upon local charge accumulations, despite the reasonably large probing depth. We have added the following discussion into the revised manuscript in Page 12 to clearly address this point:

“The probing depth of EFM, to first order, is roughly proportional to the product of the dielectric constant (ϵ) and the tip-sample distance (z), i.e. $\sim\epsilon z$ ^{50,51}. Taking the dielectric constant of HfO_2 (~ 25)⁵² and the present experimental parameters ($z = 20$ nm) into

consideration, the probing depth can theoretically reach ~500 nm. However, in practice, the thickness of the HfO₂ film is only ~5 nm in the present study, and beneath HfO₂ the electric field will be fully screened by the metallic TiN electrode. As a result, in the present system the effective probing depth is limited by the dielectric thickness. It should be pointed out that although the reasonably large probing depth endows EFM with subsurface characterization capability⁵⁰, meaningful EFM signal will only be collected when electrostatic force arises, e.g. upon local charge accumulations, as in the case of Figs. 1–2 and Supplementary Fig. 4.”

The second part of the question is regarding the potential applicability of the proposed technique to lateral thin film devices. In order to verify this, we have prepared new lateral devices using e-beam lithography, as shown in Supplementary Fig. 5a. The device has a nominal electrode distance of ~150 nm. By applying a positive voltage sweep on the right electrode with respect to the left one (Supplementary Fig. 5b), theoretically oxygen ions are expected to accumulate around the anode. On the contrary, as also pointed out by reviewer #2 (question 1), oxygen vacancies originally existing in the as-prepared oxide film should get accumulated at the counter electrode at the same time. Supplementary Fig. 5c clearly demonstrates the existences of charge accumulations at the surfaces of both electrodes, as indicated by the dashed lines. However, these may not be directly related to oxygen ions/vacancies, since both the electrodes are present and thus form a capacitor, unlike the previous HfO₂/TiN samples. The charges stored in the capacitor due to preceding voltage applications may complicate the interpretation and assignment of the electrostatic signals, while removal of the electrodes may be a destructive process. Nevertheless, these results unambiguously demonstrate the proposed technique in the present study indeed has the capability of analyzing lateral thin film devices, in addition to the vertical devices used in the original manuscript, although the capacitance effect needs to be carefully decoupled from the ion transport. We therefore sincerely thank the reviewer for giving this important advice.

We have included these new data in Supplementary Information, along with further discussions. We also added the following discussion in Page 13 of the main text to clearly address this point: *“It is also worthwhile noting that the proposed approach is applicable to the characterization of ion motion in lateral devices as well in addition to the above devices in*

vertical configuration, if the capacitance effect can be safely decoupled (see Supplementary Fig. 5, Supplementary Note 5)”.

Supplementary Figure 5 | Electrostatic force microscopy for probing ion motion in lateral devices. (a) Optical micrograph of the lateral device fabricated by e-beam lithography. The arrow indicates the device with a nominal gap size of 150 nm, as enlarged in (b). (b–d) Topographic (b), 1ω (c) and 2ω (d) measurements on the lateral device in (a) with a positive voltage sweep applied on the right electrode in preceding C-AFM measurement.

3. For the O₂ bubble part (Fig. 5), since it has been previously reported and well understood by the community, I’ll suggest to move this part to the Supplementary information.

Our response: Thank you for the advice. Following the reviewer’s suggestion, Fig. 5 in the original manuscript has been moved to the Supplementary Information as Supplementary Fig. 6 in the revised version.

4. There are few typos in the manuscript. For example, on Page 8, “Fig. 11” should be “Fig. 1i”.

Our response: We would like to thank the reviewer for carefully reading our manuscript and raising this very detailed comment. We have checked the figure number in Page 8 as mentioned by the reviewer, which refers to the figure panel showing the I - V curve of the HfO₂/TiN sample under an applied voltage of 0–8 V. In fact, the figure number that we used in the manuscript was not “Fig. 11” that the reviewer mentioned, but was actually “Fig. 1l”. We understand the confusion might originate from the similar appearance between lowercase English character “l” and the number “1”. The figure number originally used was in agreement with the actual labeling in Fig. 1, but we do appreciate the very careful review from the referee.

Besides, we have also thoroughly checked the whole manuscript to remove any spelling mistakes.

Comments from Reviewer #2

Overall Remarks: The manuscript by Yuchao Yang et al. reports on AFM-based studies combine with dielectric spectroscopy measurements and TEM/EDX analysis of VCM cells. HfO₂, Ta₂O₅ and Al₂O₃ oxides were used as variety of the solid electrolyte materials with different transport properties. The authors combined and amended several techniques to verify the oxygen ion dynamics within the studied oxides. I find particularly interesting the way of distinguishing oxygen ions from electrons by electrostatic measurements.

The manuscript is well structured and written. The authors suggest a nice idea and practically demonstrated its application. The topic is of principle interest and worth of publishing in Nature Communications.

Our response: We would like to sincerely thank the reviewer for pointing out the novelty and significance of our study, and also for raising very interesting and important questions in his/her review. We have carefully considered the reviewer's advices, performed extensive new experiments and revised the manuscript to addressed all the questions, as can be found below.

1. The authors have confirmed that oxygen ions are attracted to the surface of the oxide at positive bias. This implies that oxygen vacancies are attracted at the other (negative) electrode. Reaching particular voltage, oxygen molecules will be formed at the positive electrode (as also discussed by the authors). The formed vacancies will diffuse to the cathode and new oxygen ions will come to be oxidized at the anode. If the vacancies cannot move away the oxygen ion transport (and therefore further reaction) will also be stopped. The counter electrode reaction should be reduction of the oxide (oxygen vacancies + electrons) at the cathode. This is a classical situation known from the solid state ion conductors, where a virtual cathode (as defined for the VCM cells) is formed from a reduced oxide. However, Fig 7 shows some inconsistence with this picture. Fig 7d shows (correctly) the accumulation of oxygen ions, but no accumulation of oxygen vacancies at the negative electrode is shown. Further Fig 7 (e-g) shows that the filament (reduced oxide)

grows from the anode, whereas it should grow from the cathode (to form the filament as a virtual cathode). This point should be corrected in the figure.

Our response: We would like to greatly thank the reviewer for these detailed and valuable suggestions on improving the figure to show more exact filament growth dynamics. In light of this advice, we have re-drawn the figure to reflect correct implications on the switching dynamics of VCM (now Fig. 6 in the revised manuscript, since the original Fig. 5 has been moved to Supplementary Information as suggested by reviewer #1). The modifications include: 1) oxygen vacancy accumulation at the bottom interface is clearly illustrated in Fig. 6d, in addition to oxygen ion accumulation at the top interface; 2) Figs. 6e–g now clearly depict the filament growth process from the cathode, as a result of continuous oxygen vacancy migration and counter electrode reactions.

Figure 6 | Reversible oxygen ion dynamics in forming and reset processes. (a–c) Topographic (a), 1ω (b) and 2ω (c) measurements on the HfO_2/TiN sample after positive voltage sweepings to 3, 5, 7, and 9 V at different locations in the upper row, along with

bidirectional voltage sweepings of (5 V, 0 V), (5 V, -1 V), (5 V, -3 V) and (5 V, -5 V) in the lower row. Scale bar: 3 μm . **(d–i)** Schematic illustration of dynamic ion motion and filament formation/dissolution processes during the forming and reset processes of HfO_2 memristors. **(d)** A positive voltage on the Pt tip attracts oxygen ions to the top interface and repels oxygen vacancies to the bottom interface. **(e)** Oxidation of the oxygen ions at the top interface leads to oxygen gas eruption and oxygen vacancies formation. **(f)** The migration of oxygen vacancies to the bottom interface and continued redox reactions lead to gradual growth of the oxygen vacancy filament. **(g)** Formation of an oxygen-deficient filament through the HfO_2 film. The device is switched to on state in this case. **(h)** Reversal of applied voltage will drive oxygen ions away from the top interface and oxygen vacancies away from the bottom interface. **(i)** Continued application of negative voltage on the tip leads to dissolution of the oxygen-deficient filament. The device is switched to off state in this case.

2. The authors observed also morphological changes (craters) beneath the AFM-tip at higher voltages. Can the authors exclude some interaction (maybe alloying) between the tip and the oxide at these extreme conditions?

Our response: Thanks a lot for raising this very interesting question. In order to figure out if there exists any tip-oxide interaction like alloy formation when the volcanos/craters are created, we have performed extensive new experiments including EDS mapping and spectral analysis as well as Electron Probe Microanalyzer (EPMA) characterization and Wavelength Dispersive X-ray Spectroscopy (WDS) analysis. Moreover, in order to fully examine this possibility, both the crater itself and the Pt probe used to form the crater were examined. The results are displayed in Supplementary Fig. 9, as also appended below for the reviewer's convenience.

Supplementary Figure 9a shows a typical crater that is formed on the HfO_2/TiN sample, whose stacking sequence in vertical direction is illustrated in Supplementary Fig. 9b. All the elements included in this stacking structure, i.e. Si, Ti, N, Hf and O, were detected in the EDS measurement, however, with no signal indicating existence of Pt (Supplementary Fig. 9c). If alloying occurs between the Pt tip and the HfO_2/TiN sample in forming process, Pt element

should get incorporated into the sample, at least in the crater region, which is in contrast to the above experimental results. We have also performed EDS mapping on the Pt L edge, and once again only background noise was collected, as shown in Supplementary Fig. 9d.

In addition to the above characterization on the crater, a complete examination on the alloying possibility should also include analysis on the Pt tip, as shown in Supplementary Figs. 9e–f. It is expected that alloy formation should lead to cross contamination to the tip as well. *It is interesting to find out that the end of the Pt tip seems to have gone through a melting process in C-AFM and a sphere-like shape was formed therein, probably due to the accompanying significant Joule heating effects during electroforming.* In order to further check on the possibility of alloy formation, two areas labeled as “A” and “B” were subjected to EDS analysis. While region A is pristine Pt tip, region B was in direct contact with the HfO₂ surface during the C-AFM measurements, which has been identified by the existence of a small flat surface in region B due to the tip-sample contact. Supplementary Fig. 9g shows that both regions are very similar in composition and only Pt can be detected (besides C and O that are well-known contamination elements from the ambient), while characteristic peaks from Hf or Ti cannot be observed, once again ruling out the possibility of alloying.

These conclusions were further testified by WDS analysis in EPMA, whose sensitivity in compositional analysis is about one order of magnitude higher than that of EDS, however at the cost of spatial resolution (Ref. R8). Supplementary Fig. 9h shows the analyzed region by EPMA-WDS, where region B in Supplementary Fig. 9f has been included in the analyzed region, and the results from different types of crystals and channels once again exclude the existence of Hf or Ti on the Pt probe (Supplementary Fig. 9i). Unfortunately, such analysis could not be conducted on the nanoscale craters (~500 nm in diameter), due to the spatial resolution limitation of EPMA as mentioned above.

We believe based on the extensive analysis shown above, it can be safely concluded that due to the extreme programming conditions during the forming process, *physical* interactions do occur between the Pt tip and the HfO₂/TiN sample, mainly manifesting as Joule heating effects as suggested by the melted end of the Pt tip, which is also consistent with our TEM observations as discussed in the manuscript (Figs. 5b–g). However, *chemical* interactions

between the probe and the sample, such as alloy formation, seem to be absent.

In order to clarify this point, we have added these new results as Supplementary Fig. 9, along with Supplementary Note 9. We also added the following sentences into Page 16 of the main text to reflect the correct implication: “*Such thermal effects lead to physical interactions between the probe and the sample as confirmed by comprehensive material characterizations, where it was found that the end of the Pt tip used in C-AFM can be melted during the “volcano” formation process (see Supplementary Fig. 9, Supplementary Note 9)*”.

Supplementary Figure 9 | Characterization of tip-sample interactions. (a) SEM image of a typical crater that is formed on the HfO₂/TiN sample, whose stacking structure is illustrated in (b). (c) EDS spectrum from the crater region shown in (a), with no signal shown for Pt. (d) EDS mapping for the Pt L edge in the same region shown in (a). (e) SEM image of the Pt tip that was used to create the crater shown in (a), and a zoomed-in image is shown in (f). (g) EDS spectra from the regions “A” and “B” in (f) as indicated by rectangles. (h) EPMA image of the Pt tip that was used to create the crater, depicting the region for WDS analysis, and (i) corresponding WDS spectra from different crystals.

3. The electrical measurements have been performed at ambient conditions. Could the authors exclude effects from moisture? There are several reports on effects of moisture on the switching behavior of Ta₂O₅ and Al₂O₃.

Our response: Thanks for the constructive comment. We fully agree with the reviewer that moisture is an important factor in the switching behavior of memristive devices, and it is therefore important to examine the influence of moisture on the switching behavior of our samples. To this end, we have performed additional experiments on the HfO₂/TiN samples using C-AFM in two different environments: i) *in air (with moisture, the moisture level was controlled to be ~35%)* and ii) *in vacuum (with no or limited moisture)*. The vacuum level of the SPM system (Scienta Omicron VT-SPM) used in the new experiments was $\sim 8 \times 10^{-11}$ mbar, and the sample was also baked at ~ 140 °C before measurements in order to further remove the absorbed moisture. The ultrahigh vacuum SPM setup used and the experimental results in both air and vacuum are shown in Supplementary Fig. 7, as appended below for the reviewer’s convenience.

Indeed, as shown in Supplementary Fig. 7b, when the same bias voltage of 6 V was applied on the C-AFM tip, the sample was always switched to on state in less than 100 s in air, as confirmed by >10 repetitive tests. In stark contrast, repetitive measurements have shown that successful switching was not achieved on the same sample in vacuum condition in much longer time, i.e. up to 800 s. These results clearly demonstrate the role of moisture in facilitating

resistance switching, in agreement with previous reports on the effect of moisture for both VCM and ECM devices (Refs. R1–R4).

In addition to the above current–time ($I-t$) measurements employing fixed voltage biases, current–voltage ($I-V$) measurements have suggested similar role of moisture in resistance switching. The HfO₂/TiN sample was switched to on state at around ~8.4 V in air (Fig. 1p), however the switching voltage has increased to ~10 V in vacuum, as shown in Supplementary Fig. 4j and discussed in the response to question 1 of reviewer #1. These results are once again consistent with the role of moisture in facilitating resistance switching processes.

As a result, we have included the additional experimental data in Supplementary Information as Supplementary Fig. 7 and added related discussion in Supplementary Note 7 and page 14 of the main manuscript to reveal the correct implication on the moisture effect:

“It should also be noted that ambient conditions, in particular moisture levels, were found to play a significant role in filament formation (see Supplementary Fig. 7, Supplementary Note 7), as verified by the easier programming of the same devices in air (with moisture) compared with that in high vacuum (with no or limited moisture). This is in agreement with previous studies on the effect of moisture in both VCM and ECM devices^{54–56}.”

Supplementary Figure 7 | Influence of moisture on resistance switching. (a) Experimental setup used for SPM study in the vacuum ($\sim 8 \times 10^{-11}$ mbar). (b) Comparison of the switching time in air and in vacuum when the same voltage of 6 V was applied on the HfO₂/TiN sample.

4. Figure 6 should be improved. It is difficult to justify the results now. From the shown images it is almost impossible to distinguish any particular feature in the element distribution analysis. The size of the images should be enlarged.

Our response: We would like to thank the reviewer for the kind advice on improving the presentation of the figure. To clearly distinguish the conduction channel in the elemental distribution, we have used the DigitalMicrograph software (Gatan inc.) to analyze the O K edge mapping results in the HfO₂ layer (Fig. 5j, since the original Fig. 5 has been moved to supplementary information as suggested by reviewer #1), and a histogram corresponding to the spatial intensity of the oxygen signal in HfO₂ was obtained (in log scale), as shown in Fig. 5k. The histogram clearly shows the existence of an oxygen-deficient channel on the right side of the HfO₂ layer. This was further confirmed by the line profile analysis on the O K edge conducted in STEM (Fig. 5h), where a dip in oxygen concentration was experimentally observed in exactly the same region. These well-matching results therefore justify the formation of an oxygen-deficient conducting filament in the HfO₂ sample.

Besides, following the reviewer's suggestion all the figure panels have been thoroughly enlarged and re-organized to exhibit the results more clearly. We have also marked the contour of the filament region in both the O K edge mapping (Fig. 5j) as well as the overlaid mapping results (Fig. 5o) using dashed lines, in order to provide a guide to the eyes. Related discussions in Page 16 of the main text have also been modified accordingly.

We also feel it is worthwhile noting that such difficulties in identifying oxygen-deficient conducting filaments using TEM/STEM imaging and elemental analysis actually highlight the significance of resolving oxygen ion motion and oxygen vacancy based filaments by taking advantage of the charge attribute of ions, as proposed in the present study. We believe the capability of resolving reversible and dynamic ion motion using EFM demonstrated in this work could provide a generalized approach for investigations of detailed ion transport in various memristive systems and electrolyte materials, especially for light elements.

Once again we want to thank the reviewer for the kind suggestion. The revised figure is shown as follows:

Figure 5 | Direct identification of oxygen-deficient conducting filament in HfO₂. (a) SEM image showing a collection of electrically stimulated local regions using C-AFM. The regions were set into different controlled resistance states by applying varied programming conditions. The arrows indicate regions undergoing severe structural deformations by applying strong programming conditions, while the solid red and dotted white circles suggest regions existing at pristine and intermediate switching states by applying weak and moderate programming conditions, respectively. Scale bar: 2 μm . (b–f) EDS mapping of Pt M, O K, Hf M, N K and Ti K edges and (g) overlaid mapping results in a “volcano” region with structural deformations. Scale bar: 100 nm. (h) HAADF STEM image overlaid with O K line profile and EDS mapping, showing formation of an oxygen-deficient filament in HfO₂. Scale bar: 10 nm. (i–n) EDS mapping of Pt M, O K, N K, Ti K, Hf M edges and (o) overlaid mapping results for the same region shown in (h). The contour of the filament region was indicated by the dashed lines and arrows. The histogram depicting the spatial intensity of O K edge signal in the HfO₂ film is extracted and shown in (k), highlighting the existence of an oxygen-deficient conduction

channel on the right side of the HfO₂ film. Scale bar: 10 nm.

References

- R1. Lübben, M., Karakolis, P., Ioannou-Souglideridis, V., Normand, P., Dimitrakis, P. & Valov, I. Graphene-modified interface controls transition from VCM to ECM switching modes in Ta/TaO_x based memristive devices. *Adv. Mater.* **27**, 6202–6207 (2015).
- R2. Messerschmitt, F., Kubicek, M. & Rupp, J. L. M. How does moisture affect the physical property of memristance for anionic-electronic resistive switching memories? *Adv. Funct. Mater.* **25**, 5117–5125 (2015).
- R3. Tappertzhofen, S., Valov, I., Tsuruoka, T., Hasegawa, T., Waser, R. & Aono, M. Generic relevance of counter charges for cation-based nanoscale resistive switching memories. *ACS Nano* **7**, 6396–6402 (2013).
- R4. Tsuruoka, T., Terabe, K., Hasegawa, T., Valov, I., Waser, R. & Aono, M. Effects of moisture on the switching characteristics of oxide-based, gapless-type atomic switches. *Adv. Funct. Mater.* **22**, 70–77 (2012).
- R5. Zhao, M., Gu, X., Lowther, S. E., Park, C., Jean, Y. C. & Nguyen, T. Subsurface characterization of carbon nanotubes in polymer composites via quantitative electric force microscopy. *Nanotechnol.* **21**, 225702 (2010).
- R6. Arinero, R., Riedel, C. & Guasch, C. Numerical simulations of electrostatic interactions between an atomic force microscopy tip and a dielectric sample in presence of buried nano-particles. *J. Appl. Phys.* **112**, 114313 (2012).
- R7. Robertson, J. High dielectric constant oxides. *Eur. Phys. J. Appl. Phys.* **28**, 265–291 (2004).

R8. Laigo, J., Christien, F., Le Gall, R., Tancret, F. & Furtado, J. SEM, EDS, EPMA-WDS and EBSD characterization of carbides in HP type heat resistant alloys. *Mater. Charact.* **28**, 265–291 (2004).

REVIEWERS' COMMENTS:

Reviewer #1 (Remarks to the Author):

The authors have added new results and addressed all my questions. I think the current manuscript should be suitable to be published on Nature Communications.

Reviewer #2 (Remarks to the Author):

In their point-by-point responses the authors have addressed the comments and modified/amended the manuscript.

Additional studies have been performed to verify whether alloys are formed or not at the tip/oxide contact and found that no such interaction can be observed.

The conclusions on effects of ambient/moisture are also supported by additional experiments in vacuum.

In my opinion the revised manuscript can be accepted for publication.